# Cardiac Imaging and Management of Cardiac Disease in Asymptomatic Renal Transplant Candidates: A Current Update

**DOI:** 10.3390/diagnostics12102332

**Published:** 2022-09-27

**Authors:** Eirini Lioudaki, Ariadni Androvitsanea, Ioannis Petrakis, Constantinos Bakogiannis, Emmanuel Androulakis

**Affiliations:** 1Renal Unit, King’s College Hospital NHS Foundation Trust, Denmark Hill, London SE5 9RS, UK; 2Department of Nephrology, Friedrich-Alexander University Hospital, 91054 Erlangen, Germany; 3Department of Nephrology, University of Crete, 700 13 Heraklion, Greece; 4Third Cardiology Department, Aristotle University of Thessaloniki, 541 24 Thessaloniki, Greece; 5Imaging Centre, Royal Brompton and Harefield Hospitals, Guys and St Thomas’ NHS Foundation Trust, London SW3 6NP, UK

**Keywords:** renal transplantation, cardiovascular screening, timing, coronary angiography, end-stage kidney disease, chronic kidney disease

## Abstract

Given the high cardiovascular risk accompanying end-stage kidney disease, it would be of paramount importance for the clinical nephrologist to know which screening method(s) identify high-risk patients and whether screening asymptomatic transplant candidates effectively reduces cardiovascular risk in the perioperative setting as well as in the longer term. Within this review, key studies concerning the above questions are reported and critically analyzed. The lack of unified screening criteria and of a prognostically sufficient screening cardiovascular effect for renal transplant candidates sets the foundation for a personalized patient approach in the near future and highlights the need for well-designed studies to produce robust evidence which will address the above questions.

## 1. Introduction

Cardiovascular disease (CVD) is the leading cause of death among patients with chronic kidney disease (CKD), with risk increasing with advancing stage and patients with end-stage renal disease (ESRD) having 10–20 times higher cardiac mortality compared to the general population [1,2]. Apart from atherosclerotic coronary artery disease (CAD), impaired ventricular function is also more prevalent among patients with ESRD than the general population and is associated with worse outcomes. Dysrhythmias, valvular heart disease and pulmonary hypertension are also more common in patients with advanced CKD [3].

Renal transplantation is the renal replacement modality of choice for suitable ESRD patients, with suitability determined mainly by perioperative risk and long-term outcomes. Along with all risks associated with major surgery in high cardiovascular risk patients, renal transplantation surgery is also followed by a reperfusion “hit”, which might account for the fact that perioperative cardiac complications mainly occur within the early postoperative period [4,5]. Cardiac disease remains the main cause of death after transplantation [6,7], with pre-transplantation cardiovascular risk factors such as diabetic nephropathy and a history of CVD events predicting post-transplant events and mortality [8].

Within this context, screening for cardiac disease has been an integral part of pre-transplant assessment. However, there is a wide variation in recommendations among guidelines with regards to the target-patient population and screening investigations, which results in varying practices across transplant centers. In the present review, we will discuss the relevant guidance from international bodies, evidence on the efficacy of pre-transplant cardiac screening practices with a focus on imaging in improving early and late post-transplant cardiovascular outcomes, as well as the impact of intervention on prognosis.

## 2. Guidelines

One of the main challenges is the selection of patients who would benefit from more extensive screening while avoiding over-investigating the associated risks and costs. Following the initial assessment with history, physical examination and electrocardiogram (ECG) that all organizations concur to, guidance is less clear about which patient cohort should be further investigated in the absence of symptoms and any initial positive findings. Asymptomatic CAD (defined as at least one coronary artery with 50% or greater stenosis) has been reported in up to 53% of patients with CKD [9]. A significant proportion of ESRD patients with CVD maybe be asymptomatic or have atypical symptoms such as dyspnea even during acute events, with only 44.4% with acute myocardial infarction presenting chest pain as opposed to 72% in patients without CKD [10].

The Kidney Disease: Improving Global Outcomes (KDIGO) clinical practice guideline on the evaluation and management of candidates for kidney transplantation, which was published in April 2020, suggested that following initial assessment, patients without signs or symptoms of CVD but at high risk for CAD (e.g., diabetes, previous CAD), or with poor functional capacity, should undergo non-invasive CAD screening. It is then recommended that asymptomatic candidates with known CAD should not be revascularized solely to reduce perioperative cardiac events [11].

Along the same lines, the European Renal Best Practice Guideline [12], issued in 2015, recommended that after the initial assessment, a standard exercise tolerance test and echocardiography should be offered to asymptomatic high-risk patients (older age, diabetes, history of CVD). Non-invasive stress imaging [myocardial perfusion or dobutamine stress echocardiography (DSE)] is recommended for patients at high risk, as well as a positive or inconclusive exercise tolerance test, which in case of evidence of ischemia should then be followed by coronary angiogram.

In a scientific statement on cardiac evaluation for liver and kidney transplant candidates published in 2012 [7], the American Heart Association and the American College of Cardiology Foundation recommend that non-invasive stress testing may be considered in the absence of active cardiac disease if three or more CAD risk factors are present, such as diabetes, previous CVD, dialysis vintage longer that one year, left ventricular hypertrophy, age greater than 60 years, smoking, hypertension and dyslipidemia. It is also recommended that assessment of left ventricular function by echocardiography is performed.

The Canadian Society of Transplantation consensus guidelines [13], dating back to 2005, again recommended screening for higher-risk individuals, while the approach of management of positive findings appears to differ from other guidance. Non-invasive testing is recommended for asymptomatic patients with a history of CAD, diabetes or at least three of the following risk factors: age > 50 years, prolonged duration of CKD, family history of CAD, smoking, dyslipidemia, BMI ≥ 30 kg/m^2^ and/or hypertension. It was also recommended that very high-risk patients should be referred to cardiology for consideration of a coronary angiogram, even with a negative non-invasive test. With regards to management, it is suggested that in the event of a CAD finding on a coronary angiogram, patients could be listed following successful intervention or if they are on appropriate medical management in the case of non-critical disease. This is in contrast with the most recent KDIGO guidance, which advises against intervention merely on the grounds of listing, based on evidence anteceding the Canadian guidelines (see “Revascularization” section).

Beyond CAD, the 2005 National Kidney Foundation’s Kidney Disease Outcomes Quality Initiative (NKF/KDOQI) guidelines on the evaluation of CVD in dialysis patients recommended that impaired ventricular function should be actively sought in patients considered for transplantation, i.e., more intensely than overall dialysis patients for whom the same criteria in the general population are recommended [14].

Finally, in a less elaborate approach, the British Transplant Society and Renal Association, in its 2017 reviewed version, suggested that in the absence of compelling evidence in favor of screening asymptomatic renal transplant candidates to prevent future cardiac events or reduce long-term mortality, investigations should be performed selectively to exclude high-risk individuals [15].

Overall, most guideline sets recommend screening “high” or “very high-risk” individuals based on the presence of various combinations of traditional (e.g., diabetes, age) and non-traditional (dialysis vintage) cardiovascular risk factors. There is little suggestion on which imaging investigations should be best utilized in screening. The most recent guidance by KDIGO points towards an individualized approach based principally on risk assessment by the clinicians.

In our experience, the decision on whom to screen and which screening method to use requires an individualized approach, focusing on identifying the transplant candidates who could benefit from cardiovascular screening whilst avoiding overdiagnosis and the associated waste of time and resources.

## 3. Role of Cardiac Imaging in Screening

Most available evidence regards the role of stress echocardiography and perfusion studies as screening tools to guide further management and specifically determine the need for invasive investigations (coronary angiography); though based on local protocols, coronary angiography may have been undertaken without preceding non-invasive screening.

Some studies have focused on the association of stress-imaging findings with the occurrence of cardiovascular events in ESRD patients whilst on the waiting list, whereas other have addressed more specifically their role in predicting early and late post-transplant cardiovascular events and prognosis (Table 1).

### 3.1. Myocardial Perfusion Studies

#### 3.1.1. Data on Renal Transplant Candidates

Galvao et al. reported on 892 renal transplant candidates on hemodialysis who were assessed for transplantation utilizing stress single-photon emission computed tomography (SPECT) myocardial perfusion scintigraphy (MPS) [27]. Patients were stratified into four risk categories—very high, high, intermediate and low—based on the presence of three, two, one or none of the following risk factors, respectively: age over 50, diabetes and history of CVD. At a median follow-up of 22 months, a positive stress test was associated with a higher risk for cardiovascular events and overall mortality only in the intermediate risk group; nevertheless, there was no evidence that these results had been adjusted for other factors (Table 1). Of note, it appeared that MPS underperformed in predicting cardiovascular events in patients with diabetes compared with the non-diabetes patient group. However, in this study patients were followed up until the occurrence of an event or until renal transplantation; therefore, these findings are relevant only in regard to the cardiovascular prognosis of hemodialysis patients whilst on the waiting list for a transplant and do not inform on the predictive value of the test post-transplant (early or late).

Consistently, within a larger cohort of renal transplant candidates (*n* = 3698), 2206 patients were evaluated with MPS and followed up for a mean of 30 months (Table 1). Reversible defects on MPS appeared to be independently related with mortality, while the strongest mortality predictor appeared to be a left ventricular ejection fraction (LVEF) equal to or less than 40%. There was no information on overall MACE. The authors reported on post-transplantation data only for patients who underwent a coronary angiogram (57 patients transplanted out of 260 who had coronary angiogram), with a very small number of perioperative events documented [18].

However, a smaller study did not yield similar results (Table 1). Renal transplant candidates (*n* = 126) on the waiting list were followed up prospectively for a mean of 26 months and assessed with at least one of the following methods: risk stratification, DSE, MPS or coronary angiography. MACE defined as sudden death, myocardial infarction, life-threatening arrhythmia, heart failure, pulmonary edema, unstable angina and myocardial revascularization occurred in 14.3% of the study population, and the only significant predictor for them appeared to be the presence of coronary artery stenosis ≥ 70% on angiography [19].

Finally, focusing on a patient population with diabetes, Welsh et al. prospectively examined the performance of stress MPS as a pre-transplant assessment investigation in a renal transplant candidate cohort (*n* = 280) with diabetes (*n* = 154 type 1; *n* = 121 type 2) in predicting MACE at a mean follow-up of four years. Of the 280 patients, fewer than 50% received a renal transplant during the observation period. MACE occurred in 29% of the study population. A positive MPS was the single predictor of angiographically diagnosed CAD, but showed a negative predictive value of 64.6%. The presence of CAD on angiography was the only predictor of MACE during follow up [25].

MPS should be interpreted with caution in renal transplant candidates. This is of importance in certain patient categories, such as in patients suffering from diabetes, in whom MPS might underdiagnose CAD.

#### 3.1.2. Data on Renal Transplant Recipients

Findings with regard to transplant recipients vary across different studies and modalities. Nevertheless, renal transplant recipients who present an increased cardiovascular risk profile experience cardiovascular events in the long term, suggesting that cardiovascular imaging can be an effective tool for those patients [38]. Doukky et al. [32] examined the performance of stress MPS in a cohort of 581 asymptomatic renal transplant recipients in relation to the number of risk factors present, as described in the American Heart Association and the American College of Cardiology Foundation (AHA/ACCF) statement [7]. At a mean follow up of 3.7 years, 18% of the study population had at least one MACE. An abnormal summed stress score (SSS) on SPECT MPS was independently predictive of long-term MACE (i.e., >30 days post transplantation) in patients with three or four risk factors, but not in patients with fewer or more than that or for early MACE [32].

However, other studies did not demonstrate a predictive role for the scans. In a cohort of 182 asymptomatic transplant recipients aged over 50 without baseline CVD, MPS was diagnostic of CAD confirmed by coronary angiography in 8.9%. Nonetheless, positive MPS findings were not predictive of cardiovascular events at 12 months post-transplant, which occurred in 15.5% of the patients, while the only independent predictor left was ventricular hypertrophy on echocardiography [29]. On a similar note, SPECT MPS was not predictive of perioperative or long-term MACE in a Finnish cohort of renal transplant recipients (*n* = 301) at a median follow-up time of 96 months [34]. In this study, MACE was present in 2.6%, 2.9% and 2.7% in the early postoperative, and 27%, 29.4% and 51.4% in the longer-term for patients with normal, mildly abnormal and severely abnormal SPECT results, respectively. Of note, 27% of patients were symptomatic prior to SPECT. Moreover, in a retrospectively studied renal transplant recipient cohort (*n* = 1460), 88.1% (*n* = 898) received CAD screening with non-invasive investigations with MPS, among which 79 were symptomatic. After a median follow-up of 2.9 years, abnormal MPS results were not associated with prognosis with regard to mortality; however, the investigators did not report on other MACE apart from death [26].

Finally, in a direct comparison of the two modalities, MPS was superior to DSE in predicting death, mortality and revascularization in a renal transplant candidate population (*n* = 229) at 8 years of follow-up [39]. However, in this study the whole cohort underwent both screening investigations without risk stratification; therefore, these findings cannot be generalized as far as the role of the two investigations in pre-transplant screening is concerned, as the diagnostic accuracy of each might well vary according to the patients’ risk characteristics.

Overall, the inhomogeneity in study design among the various studies assessing the predictive role of MPS in the incidence of MACE does not allow us to extrapolate a generalized statement with respect to the predictive role of these modalities. It seems that a positive MPS is associated with increased long-term MACE in high-risk transplant candidates.

### 3.2. Stress Echocardiography

Assessing the role of stress echocardiography (exercise and dobutamine) in pre-transplant screening, Tita et al. demonstrated that among high-risk individuals (*n* = 149, based on the presence of at least one of the following: age > 50 years, diabetes, abnormal ECG and history of angina or congestive heart failure), abnormal results were strongly and independently associated with MACE following transplantation and over a mean follow-up of 2.85 years [22]. Sixteen out of 149 patients had MACE within the first year, and stress echocardiography was found to have a low sensitivity (37.5%) and positive predictive value (33.3%), but high specificity (95.3%) and negative predictive value (96.1%) in these intermediate–high-risk patients [22]. Consistently, in a retrospective, single-center trial, the combination of fixed and inducible regional wall-motion abnormalities (RWMA) on DSE was associated with a more than five-fold risk of MACE at a mean follow-up of 48 months in a cohort of renal transplant recipients (*n* = 185) with at least one of the following factors: age ≥ 50, diabetes mellitus, previous myocardial infarction or stroke or atherosclerosis. MACE occurred in 13% of the patient population [24]. When compared with angiography, DSE presented a sensitivity of 88% for detecting significant coronary lesions.

Among high-risk individuals, positive stress echocardiography could predict long-term MACE after transplantation, more so in the high-risk subgroup.

### 3.3. Other Imaging Modalities

In the era of multimodality cardiac imaging, the role of cardiac MRI (CMR) and coronary computed tomography angiography (CTA) in pre-transplant assessment merits consideration. Patel et al. examined the survival benefit of various non-invasive and invasive screening methods in a renal transplant candidate cohort, which they followed prospectively for a median 2.6-year period. Among 300 unselected ESRD patients assessed with Bruce exercise testing (ETT) and CMR, 222 were wait-listed, of which 80 were transplanted during the follow-up period.

Positive CMR tests (the presence of a left ventricular systolic dysfunction or LVH in CMR) were not associated with mortality, while the authors did not report on other MACE. The significance of the inability to exercise in this context is likely to reflect overall frailty rather than poor cardiovascular health, especially in the absence of any link to positive stress or CMR tests with death in this study [21].

In a smaller study, which included 62 asymptomatic but high-risk for CVD renal transplant candidates, dobutamine stress MRI appeared to have 100% sensitivity and 89% specificity in diagnosing angiographically significant CAD in 62 renal transplant candidates [40].

Furthermore, the role of coronary CTA and the coronary artery calcium score (CACS) was prospectively assessed and compared to SPECT and invasive coronary angiography in 138 renal transplant candidates who underwent all of the above investigations. Coronary CTA appeared superior in detecting obstructive CAD compared to other non-invasive techniques, with a sensitivity of 93% reaching 100% when the stenosis was localized in a proximal segment and a negative predictive value of 97%. Nevertheless, it had a low specificity of 63% [41]. Both CACS and CTA were associated with MACE at 3.7 years of follow-up, while SPECT MPS was not [42]. Moreover, coronary artery calcium and epicardial adipose tissue measured by hybrid techniques such as SPECT–CT and PET–CT have been shown to correlate with abnormal MPS in kidney transplant candidates, which suggests that these newer techniques may have a role in pre-transplant cardiac assessment in the near future [43].

Key point:

Data from the implementation of the novel noninvasive diagnostic methods in the cardiovascular assessment of transplant candidates are limited. Small studies demonstrate a high accuracy of stress cardiac MRI in detecting CAD in asymptomatic high-risk transplant candidates and a potential role of coronary CTA in excluding obstructive CAD. Overall, it is necessary that these modalities are evaluated in relation to clinical outcomes in this patient population. In selected high-risk patients (especially those with increased dialysis vintage [44]), the presence of vascular calcification might further perpetuate the interpretation of CTA. In these patients, invasive methods, such endovascular ultrasound or intra-coronary pressure measurement, could be helpful in assessing the severity of coronary stenoses and therapeutic strategies.

### 3.4. Metanalyses and Propensity Score-Matched Data

The two modalities, DSE and MPS, were collectively assessed in a meta-analysis, which included 52 studies and 7401 participants. DSE and MPS were non-inferior to coronary angiography in predicting MACE and cardiovascular mortality. Nevertheless, there was not a substantial difference in the numbers of patients with positive findings on any investigation—including coronary angiography—experiencing MACE compared to those with negative results, which puts into question the role of these three investigations in this setting altogether. In a previous meta-analysis, the same group of investigators assessed the accuracy of DSE (13 studies) and MPS (nine studies) to diagnose CAD in comparison to coronary angiography in renal transplant candidates. The two tests showed moderate sensitivity and specificity for CAD, which was slightly compromised when studies utilizing a standard reference threshold of ≥70% coronary artery stenosis on angiography for CAD diagnosis were only included. Overall, DSE appeared to perform better than MPS, but as the authors concluded, this will have to be tested in direct comparisons of the two modalities [45].

Furthermore, a recently published a propensity score-matched analysis [37], which used data from the Access to Transplant and Transplant Outcome Measures (ATTOM) study, assessed the role of cardiovascular pre-transplant screening of asymptomatic transplant recipients in a five-year cardiovascular outcome risk prediction. Across 18 transplant centers in England, 2572 individuals, all transplanted between 2011 and 2017, were included in this study, of which 51% underwent some sort of CAD screening. Screening included different combinations of the following: echocardiography +/− stress test (exercise tolerance test, DSE or MPS), CT or invasive coronary angiogram. The percentage of patients screened varied between 5 and 100% across different centers. The propensity score matching based on age, sex, ethnicity, socioeconomic status, smoking history, history of diabetes, ischemic heart disease, peripheral vascular disease and cerebrovascular disease yielded two patient groups (total *n* = 1760) for comparison. MACE occurred in 0.9%, 1.3%, 2.1% 3.6% and 9.4% of the total study population at 90 days, six months, one year, two years and five years post-transplant. In the propensity-matched cohort, CAD screening with any modality was not associated with MACE at 90 days, one year or five years. Age and history of ischemic heart disease were independently associated with MACE at one and five years, while male gender was also an independent predictor of MACE at five years. In the overall study population, there was a low incidence of MACE after transplantation (0.9% at 90 days, 3.6% at one year and 9.4% at five years). Though this can be perceived as a success of screening strategies in minimizing cardiovascular events risks, the widely varying percentages of patients screened, with more or less differing protocols across centers, put this statement into question. Moreover, as the authors note, without knowledge of how many patients might have been delayed or unwarrantedly excluded from transplantation, these findings should be interpreted with caution.

The ongoing Canadian–Australasian Randomized Trial of screening kidney transplant candidates for CAD (CASRK) will investigate whether regular screening for CAD after waiting-list reduces MACE [33].

Key point:

Overall, there is no high-quality evidence to support or dispute a role for pre-transplant cardiac screening, or to recommend which would be the investigations of choice. Available data are sourced from observational cohorts and reflect a wide variation in screening strategies across centers with regard to investigations utilized, patient selection criteria, percentages of patients assessed with non-invasive and invasive methods, as well as the definition of MACE in the studies. The incidence of MACE has also ranged widely across studies, but this is most likely the result of a varying definition, from including crude events such as myocardial infarction or death to wider definitions including hospitalization for heart failure and stroke atrial fibrillation, etc. Furthermore, decisions to proceed to coronary angiography +/− revascularization have been largely based on local protocols and clinicians’ discretion, and though it is reasonable to assume that standard practices are followed, it remains unknown whether the prospect of wait-listing and transplantation have been taken into consideration during decision-making.

## 4. The Role of Revascularization

Following investigations, it remains to be determined what action should be taken in the case of positive findings. The need for revascularization has ranged widely across studies from 4.6% to 36% [29,32,34,46]. Observational studies have not shown any difference in events for those who received revascularization prior to transplant versus those who did not [21,47]. However, the lack of randomization in these studies and revascularization decisions based on local protocols render comparisons between revascularization and non-revascularization groups problematic. The best available evidence arises from randomized controlled trials examining preemptive coronary revascularization prior to vascular surgery, which so far does not support prophylactic coronary revascularization solely on the basis of reducing perioperative events. The Coronary Artery Revascularization Prophylaxis (CARP) trial randomly assigned high-risk patients (*n* = 510) with clinically significant CAD to revascularization or no revascularization before elective major vascular surgery and showed no difference in mortality between the two groups [17]. Consistently, the Dutch Echocardiographic Cardiac Risk Evaluation Applying Stress Echo (DECREASE) V showed no benefit in prophylactic coronary revascularization for patients (*n* = 101) with positive stress tests undergoing elective vascular surgery versus medical management.

Kahn et al. retrospectively examined a renal transplant cohort (*n* = 1460), 88.1% of which received non-invasive screening. It was suggested that revascularization might result in improved long-term outcomes, because patients with significant CAD managed medically had a worse prognosis. Nevertheless, patients were considered altogether, independent of the presence of symptoms, while the selection criteria of patients who underwent intervention versus medical management were not described. It is likely that a patient with severe CAD assigned to medical management rather than intervention might have been elderly, frail or in possession of other factors that render them unsuitable for intervention; therefore, it is not safe to conclude that the worse prognosis is a direct result of non-revascularization, but might well have been—at least partly—due to the patients’ background [26].

The recent ISCHEMIA–CKD trial refuted a role for intervention as a prophylactic strategy in a population with advanced CKD (*n* = 777). This was a randomized controlled trial which recruited patients with advanced CKD and moderate or severe myocardial ischemia on stress testing, and assigned them to either invasive—i.e., coronary angiography followed by revascularization if appropriate, on top of medical therapy—or only medical therapy, with invasive strategy employed if medical therapy failed. After 2.2 years of follow-up, the two groups had similar rates of the primary endpoint (composite of death and non-fatal myocardial infarction) and the key secondary outcome (a composite of death, nonfatal myocardial infarction or hospitalization for unstable angina, heart failure or resuscitated cardiac arrest) [35]. Moreover, invasive treatment was more often associated with dialysis initiation or death as well as other complications such as stroke. Subgroup analysis of the 194 transplant candidates taking part into this pivotal study also showed that the invasive approach failed to provide benefits concerning the cardiovascular burden when compared with non-invasive therapy [36].

These results are in agreement with those of the COURAGE trial, which was not limited to a CKD population. COURAGE randomized 2287 patients with myocardial ischemia on stress testing and significant coronary artery disease to receive either percutaneous coronary interventions in addition to optimal medical therapy or optimal medical therapy only. After a median follow-up period of 4.6 years, the two groups did not differ with regards to their primary outcome (a composite of death from any cause and nonfatal myocardial infarction) or any of the secondary outcomes [48].

Key point:

Overall, the decision for revascularization in asymptomatic patients with advanced CKD and myocardial ischemia on stress testing as a strategy to improve the cardiovascular post-transplant outcomes could not be supported from the previous studies.

## 5. Non-Coronary Cardiac Disease

Apart from CAD, patients with CKD are affected by various structural and functional cardiac alterations, which become more prominent with advancing stages. Uremic cardiomyopathy entails increased left ventricular mass and hypertrophy, and systolic and diastolic dysfunction and myocardial fibrosis [49], and its features, may be prevalent in up to 80 to 85% of patients with ESRD and is associated with increased mortality [50].

Heart failure is common and often underdiagnosed in ESRD [51,52], possibly due to symptoms being attributed to other renal-related factors such as renal anemia and fluid overload, and it is hence under-investigated unless in the context of screening for transplantation. Among high-risk transplant candidates evaluated with SPECT (*n* = 2718), 24.9% had an LVEF of ≤50% and 10.5% had an LVEF of ≤40% [3]. Unsurprisingly, overt heart failure is associated with worse outcomes in dialysis and transplant patients [53,54]. However, it appears that even lesser degrees of systolic dysfunction, and in the absence of symptoms, are independently associated with increased mortality in patients on the transplant waiting list [3], as well as all-cause and cardiac death and non-fatal events, independent of ischemic burden in transplant recipients [55]. Consistently, the LVEF assessed by gated myocardial perfusion imaging was the strongest predictor of mortality among renal transplant candidates (*n* = 3.698), with a 2.7% mortality increase for each 1% ejection fraction decrease [18].

Furthermore, valvular heart disease is up to five times more frequent in ESRD compared to the general population with worse outcomes, even after surgical repair [56]. Pulmonary hypertension is very common among patients with ESRD and is associated with increased mortality [57]. In a retrospective analysis of the records of patients assessed with echocardiography for transplantation, severely impaired left ventricular function, pulmonary hypertension and/or right ventricular dysfunction and RWMA were independently associated with all-cause mortality over a mean follow-up of 4.2 years. The authors combined these echocardiographic parameters with other factors predictive of all-cause mortality (age, transplant listing status and diabetes) to create a score, which when calculated appeared predictive of death according to the number of factors [58].

As a result, cardiovascular prognosis in renal transplant candidates and recipients is not solely dependent on coronary patency and risks cannot be effectively controlled by establishing improved coronary flows.

In conclusion, diagnostic and therapeutic strategies should be undertaken to improve LVEF, pulmonary hypertension and valvular diseases in renal transplant candidates.

## 6. Conclusions and Future Directions

The aging population, alongside scientific advances leading to improved quality of dialysis delivered and immunosuppression, have meant that patients who are elderly and with more complex comorbidities are increasingly considered for renal transplantation, thus comprising an expanding pool of potential transplant candidates for the years to come. Cardiac screening has been an integral part of pre-transplant evaluation, and so far it primarily revolves around coronary patency and myocardial perfusion. Guidance has evolved over time in an attempt to lessen the variation of practice across centers, but the screening processes remain far from streamlined in terms of both patient selection and appropriate investigations, especially in the absence of symptoms of CVD. The absence of high-quality data from randomized trials complicates the establishment of an optimized screening that would effectively reduce the perioperative and later post-transplant MACE risk without causing delays in wait-listing, unnecessary exclusion of patients or even harm from exhaustive and invasive investigations. So far, the evidence does not point towards stress non-invasive investigations, but it remains likely that in the absence of more solid evidence, these will continue to be employed as per local protocols and clinicians’ discretion, especially as a means to control perioperative cardiovascular risk. With data sourced from studies in vascular surgery, pre-emptive revascularization does not seem to improve the perioperative risks and needs to be undertaken as per standard indication and guidance. A more holistic approach in relation to cardiac disease to include non-coronary aspects appears particularly relevant, and it would be interesting to examine whether measuring features of uremic cardiomyopathy such as diastolic dysfunction and myocardial fibrosis might serve in risk prediction. Most importantly, randomized controlled trials assessing the role of different CAD screening methods for the individual renal transplant candidate are dutifully awaited in order to inform on current practices.

## Figures and Tables

**Table 1 diagnostics-12-02332-t001:** Studies examining the effect of various methods of screening for cardiovascular morbidity on cardiovascular outcomes in renal transplant candidates.

Reference	Screening Modality	Number of Patients	Study Population	Main Findings
[16]	Coronary angiography	110	Pre-transplant diabetes patients	Increased association of coronary artery disease with insulin-dependent diabetes after coronary angiography. No post-transplantation outcome recorded.
[17]	Coronary revascularization before elective major vascular surgery	510	Non-dialysis patients, no transplant candidates	Coronary revascularization before major vascular surgery does not alter the long-term outcome.
[18]	MPS, coronary angiography	2207 MPS, 260 coronary angiography	ESRD transplant candidates	Reduction of left ventricular ejection fraction is associated with increased mortality.
[19]	MPS, DSE, coronary angiography	186	ESRD transplant candidates	Coronary obstruction (≥70%) is the only factor associated with MACE. Low-risk stratification (absence of diabetes mellitus, peripheral arterial disease, clinical coronary artery disease).
[20]	Coronary revascularization in high-risk patients before major surgery	101	CKD patients (defined as having Cr > 1.8 mg/dL) comprise 18.9% of study population.	Coronary revascularization before major surgery does not alter the outcome.
[21]	Electrocardiogram, stress electrocardiography, cardiac MRI compared with coronary angiography	300	Single-center ESRD renal transplant candidates	Invasive investigations add little compared to non-invasive approach.
[22]	Stress echocardiography	149	ESRD renal transplant candidates	Positive study associates with increased risk for CV events. Negative study, irrespective of CV risk stratification, does not associate with CV events.
[23]	MPS	1123	Type 2 diabetes patients. ESRD patients were excluded.	CV event rates were not significantly decreased due to ischemia screening.
[24]	DSE vs coronary angiography	185	ESRD renal recipient candidates	DSE distinguish between patients with low and high CV risk.
[25]	Coronary angiography compared with myocardial perfusion study	280	ESRD renal transplant candidates. 133 patients were transplanted at a mean of 2.2 years after inclusion.	Poor negative predictive value of myocardial perfusion studies.
[26]	MPS, coronary angiography	749 (non-invasive imaging studies), 211 (coronary angiography)	ESRD renal transplant candidates-prospective after renal transplantation.	Medically managed severe coronary artery disease (≥ 70%) patients present worsened prognosis when compared with interventionally or surgically treated patients. No difference in survival between non-invasive or invasive screened patients.
[27]	MPS	892	ESRD renal transplant candidates	Associated with mortality only in intermediate-risk patients (one risk factor among age ≥ 50 years, diabetes mellitus, CVD).
[28]	Transthoracic echocardiography	343	ESRD renal transplant recipients—post-renal transplantation	Moderate and severe echocardiographic abnormalities associated with long-term CV morbidity (MACE defined as death, stroke, myocardial infarction, surgical revascularization).
[29]	Electrocardiography, echocardiography, MPS, coronary angiography	244	ESRD renal transplant recipients—post-renal transplantation	Non-invasive testing (electrocardiography, echocardiography, MPS, stress test) is proposed for high-risk patients. Risk factors associated with CV events were the presence of CVD, left ventricular hypertrophy. Absence of coronary angiography indication was protective.
[30]	Arterial stiffness measurement, echocardiography, electrocardiography	171	ESRD renal transplant recipients—post-renal transplantation	Increased arterial stiffness pre-transplantation is associated with increased CVD incidence post-transplantation. Following renal transplantation arterial stiffness is reduced.
[31]	DSE (19 studies—2689 participants), DSE (10 studies—637 participants), coronary angiography (17 studies—1947 participants)	5273	ESRD renal transplant recipients—post-renal transplantation	Non-invasive tests perform as good as coronary angiography at predicting future CV events. Negative test results do not always exclude future adverse cardiac events. Transplantation reduced the risk of all-cause mortality, CV mortality but not CV events after renal transplantation.
[32]	Myocardial perfusion study (401 patients), coronary Angiography (90 patients)	581	ESKD renal transplant recipients—post-renal transplantation	Intermediate risk (3–4 risk factors) patients benefit from myocardial perfusion study. Risk factors included age ≥ 60 years, hypertension, diabetes mellitus, CVD, dyslipidemia, smoking, dialysis > 1 year, left ventricular hypertrophy.
[33]	Non-invasive screening compared to standard of care for a given CV risk	3306	ESRD renal transplant candidates—post-renal transplantation	Recruiting
[34]	Myocardial perfusion (SPECT), coronary angiography	301	ESRD renal transplant candidates	Normal SPECT does not exclude MACE in real transplant candidates in the long term.
[35,36]	Percutaneous coronary intervention, coronary artery bypass graft surgery compared with optimal medical therapy percutaneous coronary intervention if necessary	777 (194 listed for transplant)	CKD IV, V patients	Invasive revascularization strategy is not superior to conservative management.
[37]	DSE, coronary angiography	1760	Kidney transplant recipients (prospective evaluation of DSE vs coronary angiography)	No association between screening for asymptomatic coronary artery disease and MACE when pre-transplant screening is performed.

ESRD, end-stage renal disease; CKD, chronic kidney disease; MPS, myocardial perfusion studies; DSE, dobutamine stress echocardiography; CV, cardiovascular; CVD, cardiovascular disease; SPECT, single-photon emission computed tomography; MACE, major cardiac events.

## Data Availability

Not applicable.

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
