# Peer review of "Cardiac Imaging and Management of Cardiac Disease in Asymptomatic Renal Transplant Candidates: A Current Update"

_diagnostics, 2022, doi:10.3390/diagnostics12102332_

Round 1
Reviewer 1 Report
I had the pleasure to read the present paper. Its aim is to discuss the relevant guidance from international bodies, evidence on the efficacy of pre-transplant cardiac screening practices with a focus on imaging in improving early and late post transplant cardiovascular outcomes, as well as the impact of intervention on prognosis.
The paper is written in a correct and fluent English, but I fount it a little difficult to follow.
The topic is of high interest. I think that to be suitable for publication in diagnostics, some modifications and improvement are necessary.
- First of all, I suggest to the authors at the end of each point discussed to, possibly in
cursive, make a summary of the take home message of all the articles presented. It could be interesting also to have by the authors some personal opinions concerning the point treated!
- In addition it could be useful, if present, to have the desctiprion by the authors of their personal experience, reporting some personal data. This could increase not only the impact of the paper but also the reliability.
- I suggest to cite the present papers concerning the coronaric aortic calcium score and the problem of the left ventricular hypertrophy in transplant patients:
Alfieri C, Forzenigo L, Tripodi F, Meneghini M, Regalia A, Cresseri D, Messa P. Long-term evaluation of coronary artery calcifications in kidney transplanted patients: a follow up of 5 years. Sci Rep. 2019 May 3;9(1):6869.
Alfieri C, Vettoretti S, Ruzhytska O, Gandolfo MT, Cresseri D, Campise M, Caldiroli L, Favi E, Binda V, Messa P. Vitamin D and subclinical cardiac damage in a cohort of kidney transplanted patients: a retrospective observational study. Sci Rep. 2020 Nov 5;10(1):19160.
Minor recommendations: There are several misspellings! Please correct them!
Author Response
We would like to thank Reviewer 1 for their constructive comments and feedback.
Please find a detailed response below:
- First of all, I suggest to the authors at the end of each point discussed to, possibly incursive, make a summary of the take home message of all the articles presented. It could be interesting also to have by the authors some personal opinions concerning the point treated!
We thank the reviewer for this comment. Text has now been added in each section, providing a conclusion/take home message based on data presented. We have, though, refrained from offering personal opinions based on experience for the exact reason discussed throughout the manuscript, which is varying algorithms and processes among centres and based on experience
- In addition it could be useful, if present, to have the desctiprion by the authors of their personal experience, reporting some personal data. This could increase not only the impact of the paper but also the reliability.
We thank the reviewer for this comment. As above, we will refrain from presenting personal opinions based on experience. Obviously, reporting “personal” data would require ethical approval from individual organisations of the authors, and is an individual project per se.
- I suggest to cite the present papers concerning the coronaric aortic calcium score and the problem of the left ventricular hypertrophy in transplant patients:
Alfieri C, Forzenigo L, Tripodi F, Meneghini M, Regalia A, Cresseri D, Messa P. Long-term evaluation of coronary artery calcifications in kidney transplanted patients: a follow up of 5 years. Sci Rep. 2019 May 3;9(1):6869.
Alfieri C, Vettoretti S, Ruzhytska O, Gandolfo MT, Cresseri D, Campise M, Caldiroli L, Favi E, Binda V, Messa P. Vitamin D and subclinical cardiac damage in a cohort of kidney transplanted patients: a retrospective observational study. Sci Rep. 2020 Nov 5;10(1):19160.
We thank the reviewer for this comment. We have added the citation of Alfieri et al 2019, though we have omitted the second paper by Alfieri et al. 2020, as in this study the patients were assessed merely post transplant (in contrast to the former study of the same author) which makes it appear less relevant .
Minor recommendations: There are several misspellings! Please correct them!
We thank the reviewer for this comment. We have corrected the manuscript throughout.
Reviewer 2 Report
The review describes an important clinical problem regarding cardiovascular risk in patients with CKD and after kidney transplantation. The content of the paper includes detailed descriptions of recommendations and papers describing the main topic of the manuscript. This article summarizes the current data associated with cardiac screening in patients eligible for kidney transplantation. In my opinion, the descriptions of the quoted articles are too detailed, so I recommend focusing on the study group, the intervention and conclusions in their description. Moreover, I recommend to quote the below article in the context of invasive and non-invasive cardiovascular diagnostics in future kidney transplant recipients. Assessment of cardiovascular risk during peritransplant period in renal transplant recipients. Transplant Proc. 2014 Oct;46(8):2724-8. doi: 10.1016/j.transproceed.2014.09.049. Additionally, it is necessary to correct the text in terms of spaces, commas and spelling errors (for example line no: 43, 45, 60, 76, 86, 93, 104, 109, 113, 245, 184, 192, 198, 223, 348, 351, 406, 407, 410, 411, 416, 418, 424, 431).
Author Response
We would like to thank Reviewer 2 for their constructive comments and feedback.
Please see a more detailed response below:
The review describes an important clinical problem regarding cardiovascular risk in patients with CKD and after kidney transplantation. The content of the paper includes detailed descriptions of recommendations and papers describing the main topic of the manuscript. This article summarizes the current data associated with cardiac screening in patients eligible for kidney transplantation.
In my opinion, the descriptions of the quoted articles are too detailed, so I recommend focusing on the study group, the intervention and conclusions in their description.
We thank the reviewer for this comment. We have adjusted the text throughout, to avoid too much detail, and adjust to recommended style.
Moreover, I recommend to quote the below article in the context of invasive and non-invasive cardiovascular diagnostics in future kidney transplant recipients. Assessment of cardiovascular risk during peritransplant period in renal transplant recipients. Komorowska-Jagielska K, Heleniak Z, Dębska-Ślizień A, Rutkowski B. Transplant Proc. 2014 Oct;46(8):2724-8. doi: 10.1016/j.transproceed.2014.09.049.
We thank the reviewer for this comment. The recommended paper has now been added.
Additionally, it is necessary to correct the text in terms of spaces, commas and spelling errors (for example line no: 43, 45, 60, 76, 86, 93, 104, 109, 113, 245, 184, 192, 198, 223, 348, 351, 406, 407, 410, 411, 416, 418, 424, 431).
We thank the reviewer for this comment. We have corrected the manuscript throughout
Round 2
Reviewer 2 Report
I do not have any other comments.